# Quaternary Thrusting in the Central Oman Mountains—Novel Observations and Causes: Insights from Optical Stimulate Luminescence Dating and Kinematic Fault Analyses

**Daniel Moraetis** [1,*], **Andreas Scharf** [2], **Frank Mattern** [2], **Kosmas Pavlopoulos** [3] and **Steven Forman** [4]

[1] Department of Applied Physics and Astronomy, College of Sciences, University of Sharjah, P.O. Box: 27272 Sharjah, UAE

[2] Department of Earth Sciences, College of Science, Sultan Qaboos University, P.O. Box: 36, PC 123 Al-Khod, Muscat, Oman; scharfa@squ.edu.om (A.S.); frank@squ.edu.om (F.M.)

[3] Geography and Planning Department, Sorbonne University Abu Dhabi, P.O. 38044 Abu Dhabi, UAE; kosmas.pavlopoulos@sorbonne.ae

[4] Geoluminescence Dating Research Lab., Department of Geosciences, Baylor University, Waco, TX 76798, USA; Steven_Forman@Baylor.edu

\* Correspondence: dmoraitis@sharjah.ac.ae

**Abstract:** For the first time, Quaternary thrusts are documented within the Central Oman Mountains to the northwest of the Jabal Akhdar Dome. Thrusts with a throw of up to 1.1 m displace Quaternary alluvial fan conglomerates. These conglomerates have an Optical Stimulate Luminescence (OSL) age of 159 ± 7.9 ka BP and were deposited during MIS 6 (Marine Isotope Stage). The thrusts occur in two sets. Sets 1 and 2 formed during NE/SW and NW/SE shortening, respectively. Set-1-thusts correlate with the present-day stress field of NE/SW shortening which is related to subduction in the Makran Subduction Zone, and they strike parallel to the main continuous fold axis of the Jabal Akhdar and Hawasina windows. Set-2-thrusts correspond to NW/SE shortening and Plio-Pleistocene contractional structures in the southwestern Jabal Akhdar Dome. Set-2-thrusts are probably related to local variations of the present-day stress field originating from the Musandam area which is a part of the Zagros Collision Zone. Both thrust sets mimic the main thrust directions (NW/SE and NE/SW) within the Permo-Mesozoic allochthonous units (Semail Ophiolite, Hawasina napps) of the larger study area. The investigated thrusts imply some reactivation of the Hawasina and Semail thrusts due to far-field stress either from the Makran Subduction Zone and/or the Zagros Collision Zone. The ongoing tectonic activity of this part of the Oman Mountains, which has been considered of moderate activity, is for first time identified by structural data as contractional.

**Keywords:** Neotectonics; Arabian Plate; Jabal Akhdar; thrust

## 1. Introduction

Quaternary to recent tectonic deformation in the Oman Mountains has been documented in the Southeastern Oman Mountains through uplifted marine terraces [1–3]. Moreover, seismic activity in the Dibba Fault Zone of the Musandam Peninsula and one historically recorded earthquake in the Central Oman Mountains (Nizwa area) reveal some limited but still detectable tectonic activity in northern Oman [4–6]. The Qalhat Fault near Sur (Figure 1), which belongs to a WNW-striking set of faults, seems to have triggered some recent earthquakes in the Southeastern Oman Mountains [7] (and references

therein). According to [8], several faults cut through Quaternary conglomerates. However, this study is based on remote sensing techniques without presentation of kinematic fault data. Overall, there are neither accurate kinematic data nor absolute age constraints on recently active faults. Thus, the current tectonic/neotectonic deformation of the Oman Mountains is mainly supported by deductive reasoning rather than objective data.

The proposed mechanisms for the tectonic deformation in the Oman Mountains during the Quaternary are diverse, and all of them were suggested mainly through the study of uplifted marine terraces [1–3]. A forebulge model related to the active subduction zone in the Zagros Convergence Zone area has been proposed in the eastern Arabian Peninsula by [9]. The forebulge model has been supported by [4] by studying the distribution of bajadas across northern Oman. However, the previous cited researchers rejected the idea that the forebulge in the Oman Mountains is related to the Zagros Collision Zone but rather with the Makran Subduction Zone. Moraetis et al. 2018 [8] also proposed flexural bending similarly to [9] due to forebulging as a result of ongoing convergence between Arabia and Eurasia. Hoffmann et al. 2020 [3] concluded that the observed "staircase" terrace formation in the Southeastern Oman Mountains (Tiwi area) is mainly related to the crust's volume change due to serpentinization of the underlying Semail Ophiolite. In addition, the same authors suggested isostatic response due to the weight relief from the intense karstification of the Neogene limestones [1] discussed a combination of different terrace uplift mechanisms for the mid-Miocene to Holocene, including: (1) deep rooted faults (e.g., Qalhat Fault; Figure 1), (2) forebuldge dynamics, (3) isostatic rebound, and (4) a more recent general uplift of the Arabian Peninsula possibly due to subduction slow-down of the Arabian crust in the Makran Subduction Zone.

The various theories of the neotectonics listed above may indicate that the understanding of the Quaternary and active tectonics in Oman is incomplete. More structural kinematic data and age constraints on young faults are needed to complete the Quaternary tectonic picture. For the first time, we portray fault kinematics northwest of the Jabal Akhdar Dome and apply absolute dating on the faulted country rocks to constrain the time of deformation. This will lead to a better understanding of the possibly changing stress fields, their causes and their regional presence.

After introducing the geological setting and methods, we will document field evidence for thrusts, cutting Quaternary wadi conglomerates northwest of the Jabal Akhdar Dome and northwest of Rustaq (Figure 1). Furthermore, we will present Optically Stimulated Luminescence (OSL) ages of displaced Quaternary conglomerates to time-constrain faulting. All these findings will be discussed in the context of the present-day stress field and the structures in both the autochthonous and allochthonous nappes in the westernmost tip of the Jabal Akhdar Dome. We will also discuss possible causes and mechanisms of deformation of the Quaternary tectonics of this part of the Arabian Peninsula.

## 2. Geological Setting

The Oman Mountains, including the large Jabal Akhdar Dome, formed in the course of SW-directed obduction of the Semail Ophiolite during the Late Cretaceous and later Cenozoic deformation [10,11] (Figure 1). After the ophiolite emplacement, NE-directed shearing affected the region [12]. This interval lasted from the latest Cretaceous to the Paleocene/Eocene boundary [12] and was followed by tectonic quiescence until the late Eocene [11]. Since then, the main uplift of 4–6 km of the Jabal Akhdar Dome occurred until the Miocene [11]. The Jabal Akhdar Dome is a ~70 km × 50 km large window, exposing sedimentary rocks of the Arabian Platform which are overlain and surrounded by the Semail Ophiolite and Hawasina nappes (Figure 1). Uplift was facilitated by a major extensional shear zone at the northern margin of the Jabal Akhdar Dome (Frontal Range Fault; [13]) (Figure 1). Further details on the geology and tectonics of the Oman Mountains are provided in [14].

At the site northwest of the Jabal Akhdar Dome, the only exposed postobductional rocks are Quaternary sedimentary rocks [15,16]. The northwestern end of the Jabal Akhdar Dome, in the vicinity of our study area, is marked by a slightly asymmetric anticline with a fold axis plunging shallowly towards the NW (Murri Anticline, Figure 2). This fold element can be traced into the core of and

parallel to the Hawasina Window [17,18]. The age of this fold is unknown, but the authors of [15] noted that similar oriented structures exist in Cenozoic formations of the Hadhramaut Group, southwest of the Central Oman Mountains (Figure 1). The respective structures are of post-Eocene age. Active deformation at the northern Jabal Akhdar Dome towards the sea has not yet been reported.

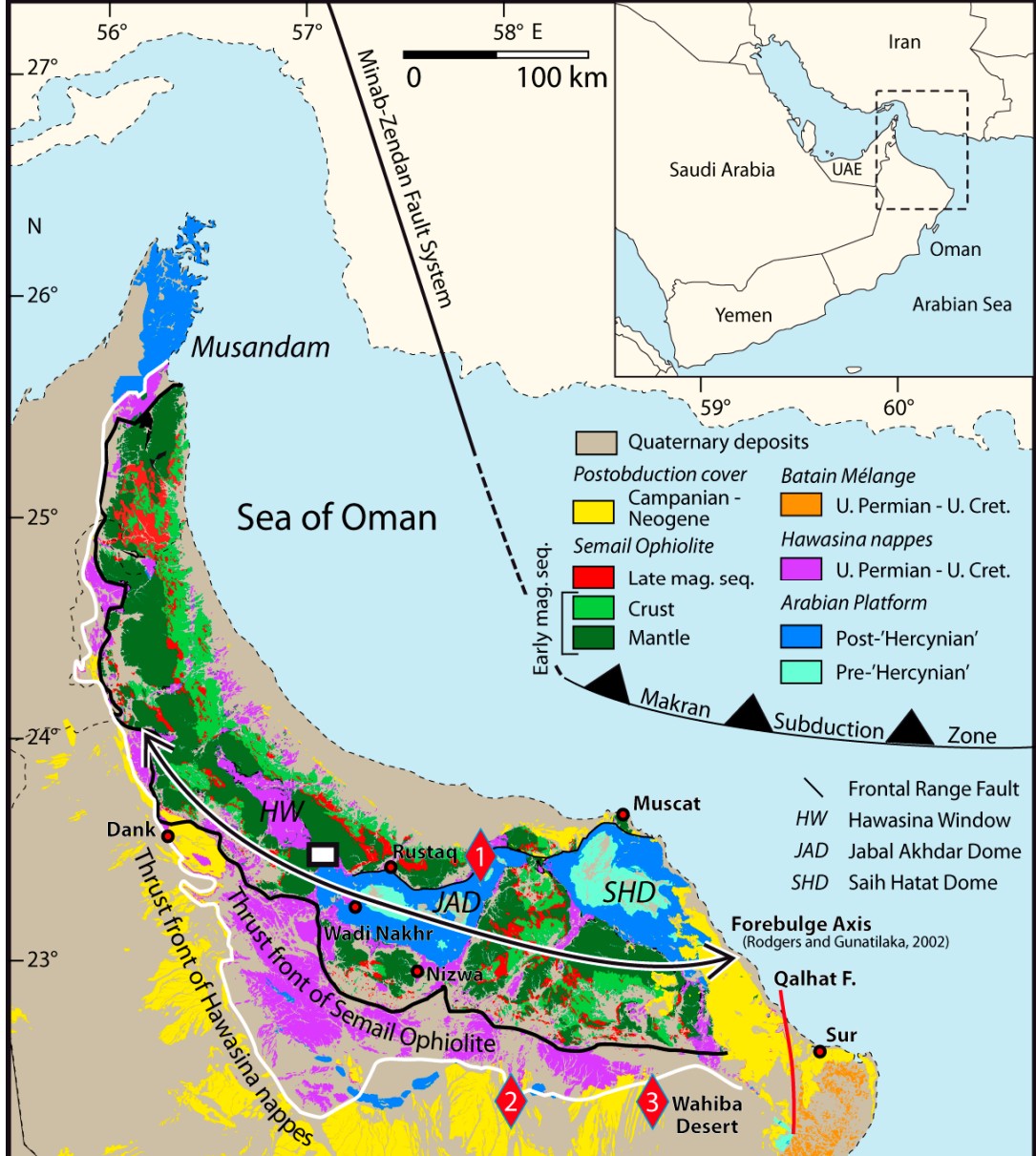

**Figure 1.** Geological overview map of the Oman Mountains, drawn after [16]. The sporadically occurring black unit below the mantle section of the Semail Ophiolite is the metamorphic sole. Study area is marked by the white rectangle. The localities Dank and Wadi Nakhr are mentioned in the text. Red rhombuses 1, 2, and 3 are the positions of alluvial fan dating from [19,20] and [21,22], respectively.

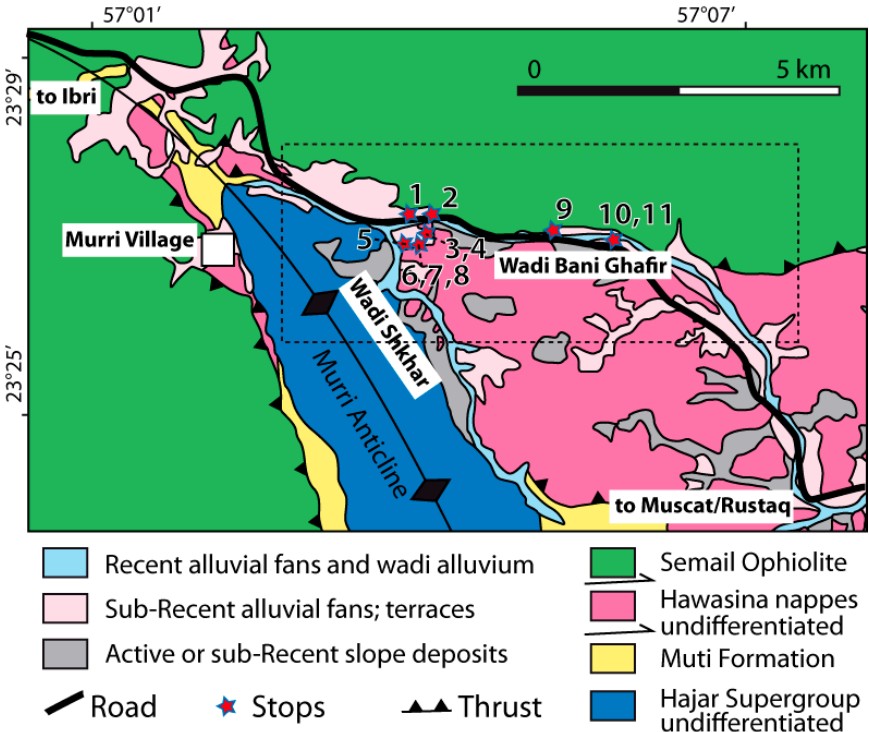

**Figure 2.** Simplified geological map of the study area, drawn after [15] (see white box in Figure 1 for location). The numbers 1 to 11 are the thrust faults as they are shown in Table 1. Optical Stimulate Luminescence (OSL) dating sample is at position next to fault 5. The rectangle marked by the black, dashed line shows the outline of Figure S2, depicting each thrust with the respective slicken lines.

## 3. Methods

### 3.1. Structural Measurements

For best accuracy we measured structural elements with a fabric compass [23]. For the kinematic analyses of fault surfaces/slickensides we used brittle shear sense indicators as listed by [24] and [25]. The coordinates and the field observations are provided in Table 1. The measured faults lie within Wadi Bani Ghafir and Wadi Shkhar (Figure 2).

### 3.2. Sampling and OSL-Dating of Conglomerate from Faulted Sediments

#### 3.2.1. Sampling for Dating

We collected a sample from Quaternary conglomerates (QUMU 5) with steel sampling tubes in two replicates (Supplementary Information). The replicates were used as one composite sample. The samples were taken ~5–6 m below the surface of the outcrop. Additionally, 1 kg of the sampled conglomerate was collected in a polyethylene bag for humidity measurement of each sample. The dating results are provided in Table 2. The sampling site was selected mainly on the basis of fine-grained conglomerate material being present, in order to be able to penetrate the deposits with the steel sample tubes. The sample locality is between the faults 3 and 4 (Figure 2). The samples were analyzed in the Geoluminenescence Dating Research Laboratory of Baylor University (Waco, TX, USA).

#### 3.2.2. Optical-Stimulated Luminescence (OSL) Dating of Quartz Grains from Fault-Related Sediments

Single aliquot regeneration (SAR) protocols [26,27] were used in this study to estimate the apparent equivalent dose of the 63–38 or 63–100 µm quartz fraction for 46 to 40 separate aliquots (Table 2). Each aliquot contained approximately 50–150 quartz grains corresponding to a 1-millimeter or less circular diameter. The analyzed sediment was compositionally and texturally immature with $SiO_2$ contents of 66–78% for the non-carbonate fraction and predominantly moderately to poorly sorted silty

sands with a variety of feldspar (Ca-rich), mafic fragments, lithic fragments and quartz. The quartz fraction of 150–250 μm was isolated by density separation using the heavy liquid Na–polytungstate, and at least one 40-min immersion in hydrofluorosilicic acid (HF 40%), to etch the outer ~10 μm of grains, which is affected by alpha radiation [28]. The aliquots purity in quartz grains was tested through infrared excitation (1.08 watts from a laser diode at 845 ± 4 nm), a visualized petrographic microscope and by Raman spectroscopy. The isolated quartz fraction was rinsed finally in HCl (10%) to remove any insoluble fluorides. The resultant final, prepared samples had a quartz purity of >99% and showed weak emissions (<400 counts/second) with infrared excitation at or close to background counts.

An Automated Risø TL/OSL–DA–15 system was used for SAR analyses. Blue light excitation (470 ± 20 nm) was from an array of 30 light-emitting diodes that deliver ~15 mW/cm$^2$ to the sample position at 90% power. Optical stimulation for all samples was completed at an elevated temperature (125 °C) using a heating rate of 5 °C/s. The fast ratio was calculated for natural emission and the equivalent emissions for a regenerative dose for each aliquot [29]. Aliquots with a fast ratio of <15 were removed from the final equivalent dose analysis; also, aliquots with infrared depletion ratio of >5%. Furthermore, a series of experiments was performed to evaluate the effect of preheating at 160, 180, 200, 220, and 240 °C on isolating the most robust time-sensitive and thermal transfer emissions of the regenerative signal prior to the application of SAR dating protocols. A test for the reproducibility of the radiation-induced SAR ratio (Lx/Tx) was also performed by giving the same beta (90Sr/90Yt) source radiation exposure for the initial and the final regenerative dose and evaluating the concordance of the SAR ratios, which should be within 10% [26,27].

The SAR protocols were used to resolve equivalent doses for QUMU 5 sample. The statistical significance of an equivalent dose population was determined for fifty to thirty-one quartz aliquots for five out of the seven samples, that were not at dose saturation (Table 2). Aliquots were removed from analysis if the fast ratio was <20 [29], the recycling ratio was not between 0.90 and 1.10, the zero dose was >5% of the natural signal or the error in equivalent dose determination was >10% [26,27]. Error analysis for equivalent dose calculations for individual quartz aliquots assumed a measurement error of 1% with 2000 Monte Carlo simulations. Recuperation was <3% for all samples, which indicates insignificant charge transfer during the measurements.

Finally, the equivalent dose (De) distributions was log normal and exhibited overdispersion values between 6% and 68% [30,31]. The environmental dose rate (Dr) was calculated through the U, Th, Rb, and K concentrations (further details are provided in the Supplementary Information in Table S1). The datum year for all OSL ages is AD 2010.

## 4. Results

### 4.1. Alluvial Fan Deposits

A sub-recent alluvial fan occupies the area between the Murri Anticline in the west and the easterly adjacent Hawasina nappes and Semail Ophiolite within Wadi Bani Ghafir (Figure 2). The deposits are polymictic with carbonate, ultramafic, and chert clasts. Clast sizes range from sand to cobbles with some boulders. The poorly developed sub-horizontal bedding is marked by alternations of conglomerate beds with different grain sizes. Furthermore, the conglomerate beds display different amounts of carbonate cement. The lower beds are highly cemented with calcite, resulting in a bright/white hue, and the thickness varies several meters (<10 m) (Figures 3b and 4, yellow arrow shows the layer with carbonate cement). The upper conglomerate unit is comparatively dark as it is unconsolidated by carbonate cement (Figure 4, white arrow shows the uncemented layer). Its thickness is less compared to the subjacent cemented conglomerate. No grain size differences have been observed between the upper and lower units of the conglomerate. The lower conglomerate unit commonly exhibits cross-bedding (Figure 3b) while in the upper conglomerate we were unable to identity such features. The two conglomerate units are probably separated by an erosional surface (Figure 4, white stripped line). Faulting had affected both units (Figure 3a, Figure 4).

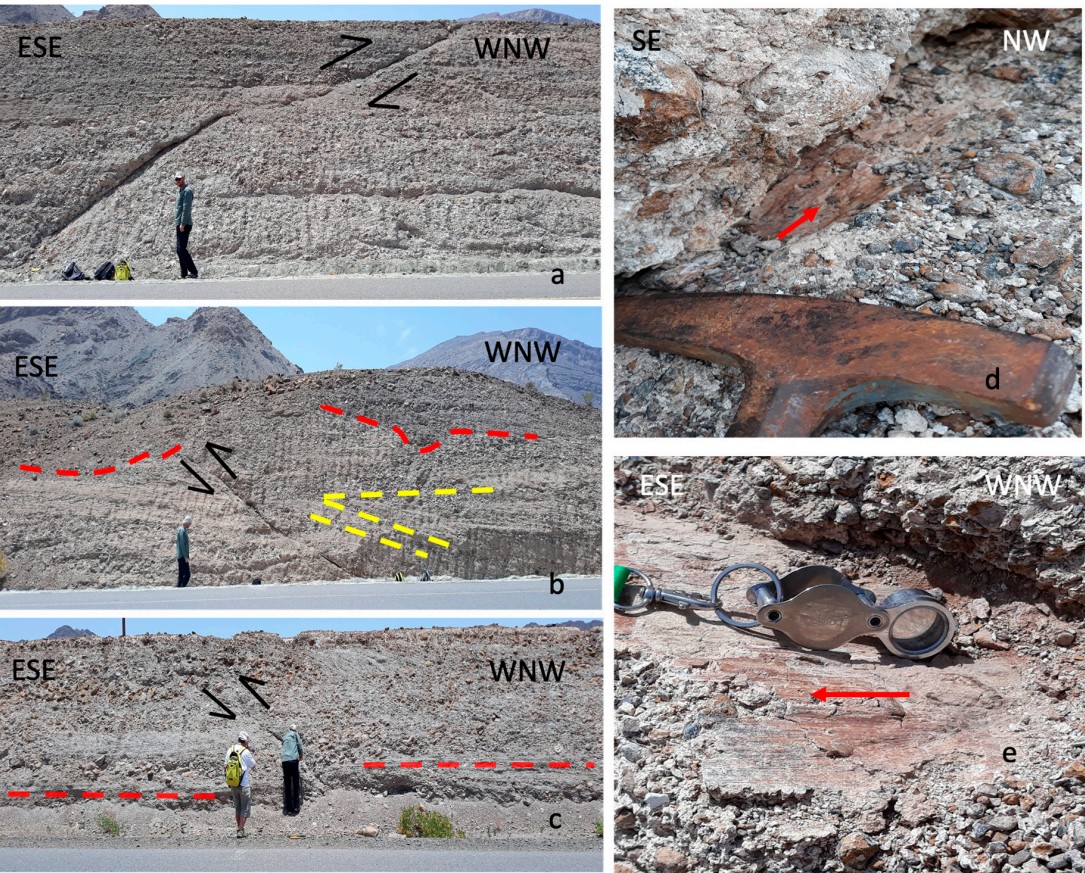

**Figure 3.** (**a**). Thrust in Quaternary deposits (fault 1), (**b**). Thrust and the two different conglomerates of the alluvial fan. The lower part is cemented and contains some cross bedding (yellow dashed lines). The upper conglomerate is unconsolidated. The contact of both conglomerates is marked by the red dashed line (fault 3), (**c**). 1-m-displacement of thrust 11. The red dashed line is a marker horizon, (**d**) and (**e**). Slickenside and slicken lines (faults 1, 3).

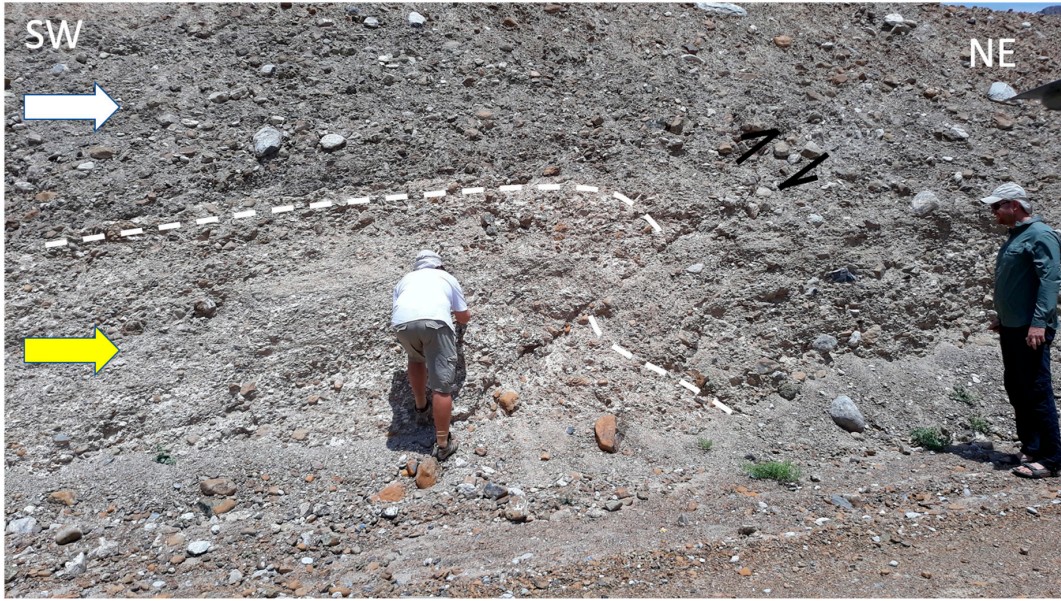

**Figure 4.** Upper unconsolidated samples (white arrow), and lower consolidated layer (yellow arrow) in fault 6. The displacement is ~1 m. Note the erosional contact between the two conglomerates (white dashed line). Also note drag folding!

### 4.2. Structural Analysis

We have identified eleven faults, six of them with dip-slip-oriented slickenlines (Figure 3, Figure 5, and Figure S2; Table 1). These faults consistently dip less than 45°. In three places, we have found displaced marker beds, revealing that the faults are thrusts. The vertical displacement of three thrusts is 1.1 m (Table 1 and Figures 3c and 4). All faults have the same simple planar morphology without ramps and flats.

The faults either strike SE/NW (set 1, six faults) or SW/NE (set 2, five faults; Figure 5 and Figure S2 in the Supplementary Information). Thrusts of each set may dip in either direction (set 1/SW and NE; set 2/SE and NW). No cross-cutting relationships between both sets could be determined. The faults cannot be traced in Google Earth because building and road constructions have modified the surface of the Quaternary hills. In addition, the identified displacement of ~1.1 m is possibly not enough to create discernible fault scarps, especially in the less cemented and friable conglomerate. We could not observe any geomorphological changes in the Quaternary hills related to the thrusts. It was also not possible to find areas of bedrock exposure below the Quaternary faults in order to identify the continuation of the faults into deeper levels.

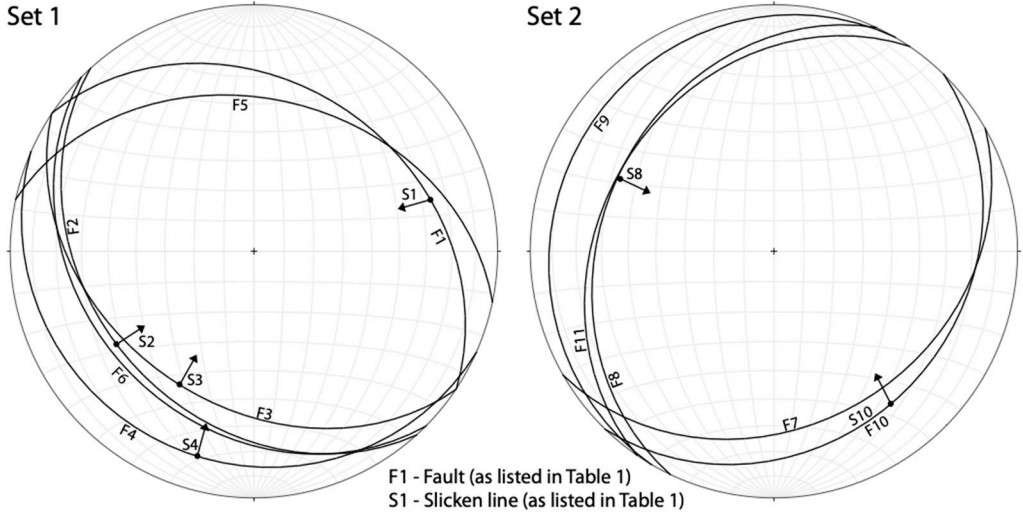

**Figure 5.** Stereonets for set-1 and set-2-thrusts (total eleven faults). These are lower hemisphere plots. Faults are depicted as circles, slickenlines are depicted as dots. Arrows indicate direction of thrust movement.

**Table 1.** Structural data for the investigated Quaternary faults. The fault numbers correspond to the stop numbers in Figure 2.

| | | Coordinates | | | Dip Direction and Dip Angle of the Plane | | Slicken Line | | Displacement |
|---|---|---|---|---|---|---|---|---|---|
| | | Degrees | Minutes | Seconds | Dip Direction | Dip Angle | Trend | Plunge | |
| QUMU-Fault1 | Lat | 23 | 27 | 41.6 | 034 | 32 | 075 | 30 | - |
| | Lon | 57 | 03 | 57.6 | | | | | |
| QUMU-Fault2 | Lat | 23 | 27 | 42.3 | 228 | 33 | 237 | 38 | - |
| | Lon | 57 | 04 | 4.4 | | | | | |
| QUMU-Fault3 | Lat | 23 | 27 | 34.9 | 214 | 38 | 210 | 35 | - |
| | Lon | 57 | 04 | 1.4 | | | | | |
| QUMU-Fault4 | Lat | Similar to fault 3 | | | 204 | 15 | 195 | 20 | - |
| | Lon | | | | | | | | |
| QUMU-Fault5 | Lat | 23 | 27 | 27.5 | 012 | 38 | - | | - |
| | Lon | 57 | 03 | 52.1 | | | | | |
| QUMU-Fault6 | Lat | 23 | 27 | 44.1 | 225 | 30 | - | | Throw 1 m |
| | Lon | 57 | 03 | 25.4 | | | | | |
| QUMU-Fault7 | Lat | Similar to fault 6 | | | 150 | 30 | - | | |
| | Lon | | | | | | | | |
| QUMU-Fault8 | Lat | Similar to fault 6 | | | 295 | 30 | 295 | 31 | Throw 1.10 m |
| | Lon | | | | | | | | |
| QUMU-Fault9 | Lat | 23 | 27 | 33.5 | 300 | 10 | - | | - |
| | Lon | 57 | 05 | 1.9 | | | | | |
| QUMU-Fault10 | Lat | 23 | 27 | 34 | 134 | 23 | 143 | 26 | - |
| | Lon | 57 | 05 | 7.1 | | | | | |
| QUMU-Fault11 | Lat | Similar to fault 10 | | | 304 | 30 | - | | Throw1.10 m |
| | Lon | | | | | | | | |

## 4.3. OSL Dating

The OSL dating results are summarized in Table 2. The age was measured within the lower part of the cemented conglomerate layer next to fault 5 (Figure 2 and Figure S1a,b in the Supplementary Information). The obtained age of 159 ± 7.9 ka BP (Before Present) (Figure 6) falls into MIS 6 (Marine Isotope Stage (MIS)). For the upper uncemented layer of the conglomerate we obtained no dating result. Our result is compared against other OSL ages from Quaternary deposits (both fluvial and alluvial) of Oman in Figure 6 and the localities of these are shown in Figure 1. A publication from Wahiba sand dune desert [21] inferred an age of 150 ± 10 ka for relict fluvial gravels and a similar age was published by [22] (Figure 6, Figure 1). A more comprehensive work from alluvial fan deposits [20] provided a large dataset from different alluvial fans from the southern flanks of the Oman Mountains with ages of different alluvial fan formation of 45 ± 5, 130 ± 19, 210 ± 19, 220 ± 24, 330 ± 34, and 410 ± 61 ka. Finally, fluvial-lacustrine sediments from northern Oman have an age of 48 ± 5 ka [19]; see Figures 1 and 6).

**Table 2.** Single Aliquot Regeneration, optically-stimulated luminescence (SAR-OSL) ages on quartz grains from a fluvial deposit (sample QUMU 5), Oman.

| Field Number | Lab Number | Aliquots [a] | Grain Size (μm) | Central/ *Minimum* Age Model De (Gy) [b] | Over-Dispersion (%) [c] | U (ppm) [d] | Th (ppm) [d] | K (%) [d] | Cosmic Dose Rate (mGray/yr) | Dose Rate (mGray/yr) [d] | Central/ Minimum Model SAR age (ka) [e] |
|---|---|---|---|---|---|---|---|---|---|---|---|
| QUMU 5 | BG4778 | 34/30 | 63–100 | 111.47 ± 4.39 | 17 ± 3 | 0.80 ± 0.01 | 1.12 ± 0.01 | 0.24 ± 0.01 | 0.23 ± 0.023 | 0.70 ± 0.02 | 159 ± 7.9 |

[a] Aliquots used in equivalent dose calculations versus original aliquots measured. [b] Equivalent dose calculated by Single Aliquot Regeneration protocols (SAR; [26,27]) Equivalent dose (De) was calculated by the Finite Mixture Model [32] and the Minimum Age Model [31]. [c] Overdispersion values reflect precision beyond instrumental error. [d] U, Th, Rb, and K content analyzed by inductively-coupled plasma-mass spectrometry by ALS Laboratories, Reno, NV; [e] includes also a cosmic dose rate calculated from parameters in [33] and it includes soft components. [e] Systematic and random errors calculated in a quadrature at one standard deviation by the Luminescence Dating and Age Calculator (LDAC) at https://www.baylor.edu/geosciences/index.php?id=962356. Datum year is AD 2010.

## 5. Discussion

### 5.1. Climatic Records and Quaternary Deposits

The polymictic character for both conglomerate units of our study area shows that the rocks of the Hajar Supergroup, the Hawasina units and the Semail Ophiolite have contributed to the clastic material. The older cemented layer indicates a wet period during its deposition. Wet periods have been deduced by speleothems in Oman which indicate pluvial events among others, during 120–135 ka BP and 180–200 ka BP [34]. Our OSL (159 ± 7.9 ka BP) age does not match the ages of [34]. However, the same age discrepancy appears in other publications, too [20–22,35]. As it has been discussed in the previous publications, recording of wet periods from U/Th stalagmites growth may not coincide with the monsoon wet maxima. Another publication [22] mentioned that pluvial events of short duration within 150 ± 10 ka BP (MIS 6) may have been left unrecorded in speleothems. Moreover, the current dating result is coeval with short pluvial events (150 ± 10 ka BP) as identified by the authors of [21,22]. In addition, MIS 6 is an interval of cool (drier) conditions which alternated with slightly warmer time periods at least between 156–165 ka BP (red arrow indicates lower $\delta^{18}O$ in Figure 6). The $\delta^{18}O$ is lowered in seawater during warm periods due to lighter $^{16}O$ release in ocean water during ice sheets melting [36] and references therein. Such alternations between warm to cold intervals within the same MIS could trigger short pluvial events. These pluvial events were the last event before the end of MIS 6 glaciation [22]. Thus, we also support a stage of fan aggradation, facilitated by possible short duration pluvial period for northern Oman, prior to the onset of aridity during MIS 5.

Hoffmann et al. (2015) [19] demonstrated the presence of a later pluvial period (in wadi deposits) for the Oman Mountains which lies within MIS 3, and a similar age has been reported also for alluvial fans from the interior of Oman [20]. Up to today, no fluvial deposits or alluvial aggradation events following MIS 6 have been detected apart from MIS 3 [20,33]. Thus, it is possible that the upper section

with the unconsolidated, uncemented conglomerate material in our area was deposited during MIS 3. A detailed future age investigation of the upper conglomerate layer would put a better constrain on both the sediment aggradation events and the Quaternary faults.

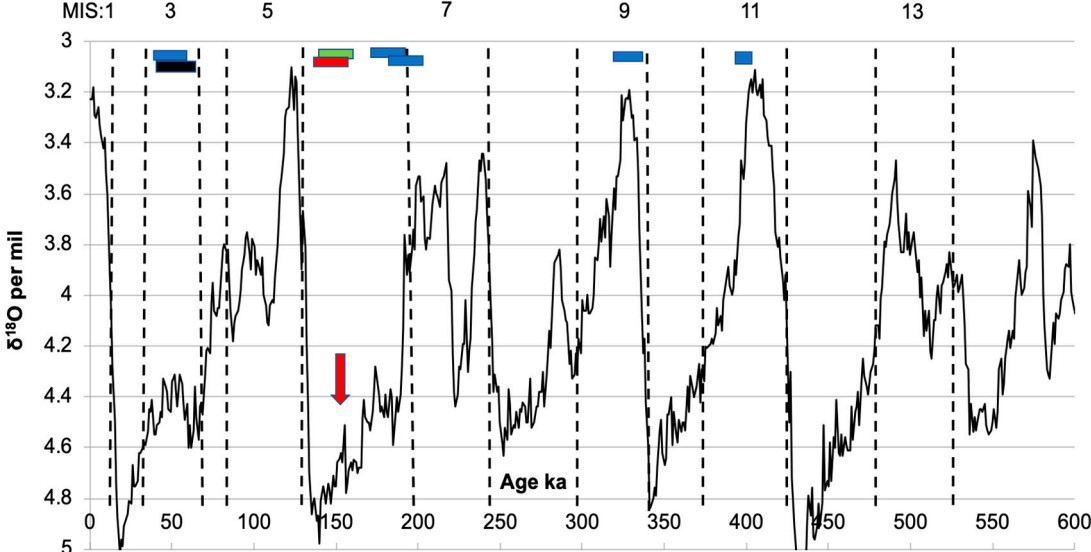

**Figure 6.** Comparison of alluvial fan and fluvial deposit ages from the interior of Oman. Every different color bar refers to different Quaternary deposits events as they have presented elsewhere. The present study age data is also included. Red bar: Ref. [21], black bar: Ref. [19], green bar: present study dating, blue bars: Ref. [20]. The LR04 stack of benthic $\delta^{18}$O from Ref. [36] was used for demonstrating MIS (Marine Isotope Stage) events. The numbers at the top of the figure indicating Marine Isotope Stages (MIS). Red arrow is showing slightly warmer period within MIS 6.

## 5.2. Structural Data and Tectonic Evolution

### 5.2.1. Set-1-Thrusts

All faults presented here are considered to be thrusts because of their low dip angle, some correspondingly displaced marker beds, drag folds (Figure 4) and their similar morphology, although not all studied faults have slickenlines and/or a shear-sense indicators. Strike-slip faults would be expected to be more or less vertical, while normal and reverse faults would be expected to mostly dip with 60–70° or >45°, respectively. The SW/NE-dipping thrusts (set 1; NE/SW-shortening) strike parallel to the Murri Anticline and the Hawasina Window (Figures 1 and 2), and they are implying a compressive stress field. NE/SW-shortening has been mentioned by several authors with an inferred time constrain of Pliocene age [37]. In addition, the same shortening direction (NE/SW) matches the present stress compression field as it is described by [38]. The present-day stress field has been attributed to convergence of the Arabian and Eurasian plates forming the Makran Subduction Zone [38]. Moreover, set-1-thrusts strike parallel to the Murri Anticline which is intriguing with respect to a possible correlation with doming of the Jabal Akhdar area. Two calcites fibers have been dated from thrusts (NNE/SSW-shortening/compression) within Cenozoic shallow-marine limestone near Dank, ~50 km to the west of our study area (Figure 1). The ages are 40 ± 0.5 and 16.1 ± 0.2 Ma [11], and the same authors attributed these ages to NE/SW-shortening and doming of the Jabal Akhdar area at the same time. The same tectonic event (NE/SW-shortening) has created other anticlines such as the Hafit Anticline during the Late Oligocene to Middle Miocene in the foreland of the Jabal Akhdar [39]. Thus, the set-1-thrusts match the Jabal Akhdar doming axis and likely indicate that some doming is still going on. A recent study along the Semail Gap at the eastern margin of the Jabal Akhdar Dome demonstrates that Quaternary conglomerates are incised by ~50 m which could be related to Quaternary uplift of the dome, neglecting climatic effects [6]. Moreover, it is important to mention

that the Semail Ophiolite and Hawasina nappes in our study area show an overall NW/SE-directed orientation of their thrust contacts (Figures S2 and S3). This orientation is in favor of a reactivation as thrusts during the present-day stress field.

### 5.2.2. Set-2-Thrusts

The set-2-thrusts (NW/SE-shortening) are not compatible with the prevailing present-day stress field. However, it has been mentioned that variations of the current NE/SW-shortening direction exist [37,38]. Such variations could be related to the opening of the Red Sea and the Gulf of Aden with a subsequent push to the Zagros Collision Zone [39]. Thus, a variation of the present-day stress field can be considered for some local NW/SE-shortening in the Arabian Plate. Another interesting fact is that the set-2-thrusts correlate geometrically (parallel) well with the thrusts of the Hawasina nappes (Figure S3) as well as with the direction of the dated structures from Wadi Nakhr, in the southwestern Jabal Akhdar Dome [40]. The four respective calcites formed between 1.6 ± 0.6 and 7.5 ± 0.9 Ma during NW/SE-shortening [40]. Thus, because of this correlation, thrusts of set 2 may have been active since the Middle Miocene up to the Pleistocene (159 ± 7.9 ka BP). NW/SE-directed compression may have contributed to the uplift history of the Central Oman Mountains. Although a cross-cutting relationship could not be established between the thrusts of sets 1 and 2, both are cutting through both conglomerate layers denoting a similar age.

Our data do not indicate that the forebulge represents the main mechanism for the neotectonic (Late Pleistocene) activity. In a forebulge, associated with uplift, extensional faults, striking parallel to the Oman Mountains would have to be expected. However, we did not observe any extensional faults in the study area. Set-1-thrusts match the present-day stress field while those of set 2 could be the manifestation of another shortening cause, possibly related to contractional forces being exerted from the Zagros Collision Zone (i.e., the Musandam area). The compressive structures in postobductional rocks of the Dank area [11], the Murri Anticline as well as the age of fractures in Wadi Nakhr [39] indicate active doming of the Central Oman Mountains.

Present-day shortening is correlated with continues doming of the Jabal Akhdar area [37,38]. However, this interpretation should be supported with more structural and age data closing the time gap between main doming of the Jabal Akhdar area (Late Eocene to Miocene) and the post-Late Pleistocene deformation presented here. Future research should focus on possible structural relationships between the Quaternary thrusts and major thrusts within the Semail Ophiolite and Hawasina nappes.

Overall, the collision of Arabia with Eurasia and/or the subduction in the Makran Subduction Zone have far-field effects, affecting the Jabal Akhdar area in the Central Oman Mountains. The Central Oman Mountains have been considered an area of moderate tectonic activity [5]. Our study demonstrates that faulting and earthquakes in this region may happen at any time.

## 6. Conclusions

The present work reveals for the first time thrusting in Quaternary alluvial fan deposits from the Central Oman Mountains. The conglomerates were deposited during the late Pleistocene (MIS 6).

Quaternary thrusting ensued in two fault sets (set 1 NE/SW-shortening; set 2 NW/SE-shortening). The first set is related to the established present-day stress field which may affect folding/doming of the western Jabal Akhdar Dome (Murri Anticline). The same shortening direction existed during the late Eocene to Miocene doming of the Jabal Akhdar area.

The thrusts of set 2 represent a regional variation of the present-day stress field, probably emanating from the Musandam area which belongs to the Zagros Collision Zone [38]. The same set-2-thrusts correspond to Plio-Pleistocene ages of compressive structures from the southwestern Jabal Akhdar Dome [40] indicating ongoing doming.

The thrusts are unlikely form during forebulging because extensional structures would be expected to have formed during this process. Future studies of a possible relationship between the thrusts of the

allochthonous Semail and Hawasina nappes with the structural data presented here could reveal that present-day shortening includes reactivation of the allochthonous nappe thrusts.

Our data demonstrate that the far-field stress of the Arabia-Eurasia collision zone can be detected in the Central Oman Mountains. This area has been considered as of moderate tectonic activity before. Our findings verify that the present-day stress field (compression) has triggered faulting and possibly past earthquakes within the Quaternary period. Thus, future faulting and earthquake occurrence are considered probable in this region.

**Supplementary Materials:** The following are available online at http://www.mdpi.com/2076-3263/10/5/166/s1, Figure S1a: Sample position for OSL analysis which is next to fault 5 (overview), Figure S1b: Sample position for OSL analysis which is next to fault 5 (detailed view), Figure S2: Detailed map of the study area, depicting different nappes within the Hawasina units and its thrust contacts, Figure S3: Schematic map depicting the thrust faults within the Hawasina nappes and the Semail Ophiolite, Table S1: Equivalent Dose (De) and Environmental dose rate (Dr) considerations.

**Author Contributions:** The study conceptualization was performed by D.M., A.S., F.M. and K.P.; methodology was constructed by, A.S., S.F. and D.M.; validation was done by D.M., A.S., F.M., K.P. and S.F.; resources provided by K.P. and S.F.; draft writing was done by D.M.; writing—review and editing was performed by F.M., A.S., S.F. and K.P. All authors have read and agreed to the published version of the manuscript.

**Funding:** This research received no external funding.

**Acknowledgments:** We are grateful for the thorough review from the reviewers and the handling of the manuscript from the editor.

**Conflicts of Interest:** The authors declare no conflict of interest.

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
