# Peer review of "Quaternary Thrusting in the Central Oman Mountains—Novel Observations and Causes: Insights from Optical Stimulate Luminescence Dating and Kinematic Fault Analyses"

_geosciences, doi:10.3390/geosciences10050166_

Round 1

Reviewer 1 Report

Overall the presented work is documenting an important finding related to the present-day stress field and ongoing tectonics in northern Oman. However, the manuscript needs to improve on consistent use of style and language as well as by optimizing some of the figures. Some of the drawn conclusions need to be extended to be more clear (especially Chapter 5.1). For details please see detailed description.

Author Response

Dear reviewer

Thank you for your efforts 

We found your comments really constructive and we include them in our revision

Answers in yellow

Reviewer 2 Report

Title: Quaternary thrusting in the Central Oman Mountains – novel observations and causes: insights from OSL dating and kinematic fault analyses

By: Daniel Moraetis, Andreas Scharf, Frank Mattern, Kosmas Pavlopoulos, and Steven Forman

The paper by Moraetis et al. is a novel contribution about the Quaternary deformation in Central Oman Mountains. They propose a compression scenario for the last 160 ka based on the kinematic analysis of only eleven thrust faults (metric to decametric-scale?) and on a single OSL date on alluvial fan conglomerate deposits. Two different thrust sets, striking NW–SE (set 1) and NE–SW (set 2), are distinguished, and slickenlines indicate NE-SW and NW-SE shortening, respectively. The results are discussed in the context of climatic records (MIS events) and the tectonic evolution of the region. Despite the low number of data, the topic is of interest for a wide audience, and I think it merits publication in the Journal. In my opinion, nevertheless, there are some aspects of the paper that could be improved in order to clarify some obscure points or to facilitate the understanding of others to an audience non-familiar with the tectonic evolution of Oman Mountains and their relationship with recent plate tectonics. There are specified in the following general comments (labelled on the revised manuscript attached to this review).

#1. Interpretation of set 1

Based on the age of deformation studied, more recent than 159 ka, I think it cannot be said that Set 1 correlate with Jabal Akhdar's main phase of doming, which took place from the early Eocene to the Miocene. I think the authors must first put it in the current context of plate tectonics (Makran subduction) and the overall present-day stress field (NE-SW shortening), and then indicate that this shortening direction was already active during the Eocene-Miocene doming phase.

#2. Interpretation of set 2

The interpretation of set 2 in the abstract and the conclusions (that is, it is the result of local variations in the present-day stress field) seem partially correct and reasonable to me, but when this point is treated in the discussion (lines 270-281) it is not in depth and it is also suggested that they are associated to far-field compresions. To delve into this aspect, it is crucial to analyze the data in relation to the largest structures located in or near the study region, in this case the Hawasina and Semail ophiolites nappes and the Murri Anticline. This analysis can give information about the possible reactivation of these structures and/or the influence of their orientation during the recent deformation. In fact, the outcrops and faults studied in this work seem to be located in Quaternary deposits located above the contact of these nappes (Figure 2). I propose to modify this figure so that these larger structures can be seen more clearly. In this regard, I have observed that the faults located in the central part (faults 1-6) have a WNW-ESE strike (N102ºE to N138ºE ), similar to that of the Semail Ophiolite contact at that point. However, the faults located to the east (faults 9-11), which have NE-SW orientations, are located in a sector where the contact of the Semail Ophiolite has changed direction to become WSW-ENE. This suggests the possibility that both the location of the described faults and the different orientation of faults and slickenlines has been controlled by the reactivation of these larger structures, especially the Semail Ophiolite nappe. In this regard, natural examples and models of different kinds show how the trajectories of the maximum principal stress sigma1 (and the resolved shear stress in the fault plane, the slickenline or transport direction) tend to become perpendicular to the ramps of the major fault planes. With the aim of adequately showing the variations in both fault orientations and their slickenlines in space and in relation to the main structures, it would be advisable for Figure 5 to also include and enlarged cartographic sketch of figure 2, with the most precise location of the measured faults, and a stereogram of each fault (of the data taken in each fault) distributed according to their position on the map (see also comment #3).

#3. Number of data and their description

Despite their low number, the examples of thrusts are compelling. In the text and in table 1, the authors indicate that 11 fault planes and 6 slickenlines have been measured (6 faults and 4 slickenlines in Set 1 and 5 faults and 2 slickenlines for Set 2). However, in the graphic representation of data for set 1 (figure 5a), 10 fault planes and 7 slickenlines are represented. I suppose this discrepancy may be due to more than one measurement being taken over several fault planes. If so, it would be convenient for all the data to also appear in the table, indicating such circumstance. With respect to figure 5, the representation of the slickenlines are not located on the fault planes in which they were measured. It would be convenient, even if manually, to associate each slickenline with its corresponding fault plane (keeping the trend of the line constant and modifying its plunge until it is on the plane). It would also be convenient to indicate the direction of movement of the thrusts in which there is information about it. Other questions that should be included in the description of the structures are: what scale are the faults? Have they been able to continue on the ground? Can these faults be mapped? Do these faults produce local changes in the geomorphology of alluvial fans such as changes in their slope, scarps ...?

#4. About OSL sampling and dating

The authors indicate that they collected two samples in the lower conglomerate unit for possible replication, but it is not clear whether the results were obtained from one sample or they used the two samples taken. This point needs to be clarified. The authors indicate that for the upper uncemented unit they did not obtain age data (line 210), which is obvious if samples for dating were not taken from this unit as can be deduced from reading section 3.2.1 (sampling for dating ). Dating the activity of thrusts would have been more accurate if this upper unit, also affected by the thrusts, had also been dated. In addition, the photos show in this unit favorable areas for sampling. It is a pity that the dating of the upper unit is not available because it would further limit the time of thrust development and also clarify some of the issues raised in the discussion, such as the speculative relationship of the deposit of this unit with the pluvial event of MIS3 (lines 243-245). Why not with the warmer and wetter MIS5?

#5. About the quite similar age of sets 1 and 2

I agree with the authors with the idea that the two thrust sets are cutting through both conglomerate layers denoting a quite similar age for them. In this regard, the authors try to ascribe each shortening direction interpreted from thrust sets 1 and 2 with the responsible stress field and its relation to plate tectonics (lines 270-281). If a similar age is assumed for them, or even if they could have been formed simultaneously, there are at least two possible interpretations that need to be discussed in the manuscript: i) that one or both sets correspond to stress deviation phenomena associated with some larger structure (see comment # 2) and ii) taking into account that the two thrust sets indicate approximately perpendicular shortening directions, that the study region was subjected to radial compression.

#6. About forebulge and uplift

It is evident that in a simple forebulge, uplift is associated with the effect of lithostatic loading on obduction. Although I do not know the structure of the region, the reading of the sources carried out during this review suggests that it is not a simple forebulge at all, since it is made up of an overlap of nappes later folded. In this regard, folding and stacking of thrust nappes, on the surface and more commonly at depth, is a common mechanism that produces uplift of the deformed region. The data provided in this work, I think, may be one of the first evidences of a slight compressive reactivation of some of the great nappes described in the Central Oman Mountains. This tectonic evolution is in clear agreement with plate tectonics data (approximation of the Arabia plate with Eurasia) and the present-day stress field data (NE-SW shortening with a strike-slip regime) described south of the Oman Mountains by Filbrandt et al. (2006).

A number of minor changes and misprint corrections habe been made directly in the revised manuscript.

Author Response

Dear Reviewer 

Your comments allow to search more and to considerably improve our manuscript

Thank you for your time and effort

Answers in yellow

Round 2

Reviewer 1 Report

Dear Authors, 

You did a fantastic job during the review process. The manuscript is now in good shape and well written; the figures improved as well.

A few minor comments are listed below. As they all relate to appearance and do not effect the scientific content, I would be fine with acceptance of the manuscript as well.

  • - Age data is still not homogenized in the text sometime written with space prior to +/-, sometimes without
  • Quotes directly used in the text are sometimes spelled out, e.g. Introduction “Kusky et al. (2005) also proposed flexural bending similarly….” But most of the time also just given as number, e.g. L253: “[22] mentioned that pluvial events of short duration within 140 to 253 160 ka BP (MIS 6) may have been left unrecorded in speleothems.”  - please homogenize in accordance to journal requirements

L83: (e.g.,[12])

L 104: Comma between the two references

L 111: “.” Missing at the end of the sentence

L183: “.” Missing at the end of the sentence  

L 192/3: “(Figures., 3, 5, 192 S2, Table 1)” remove komma and point after Figures

L 218: “.” Missing at the end of the sentence

Author Response

Thank you for your contribution. We have checked punctuation and formalities 

Reviewer 2 Report

I highly appreciate the changes made by the authors in the manuscript, which greatly improve its quality.
They should do a detailed reading and revision of the text because there are still some punctual errors (especially punctuation marks are missing / left over).

Author Response

Thank you for your elaborate work on the manuscript. We are really appreciate that 

We have gone line by line in your corrections and we standardized the manuscript